# Virus-host co-evolution under a modified nuclear genetic code

Derek J. Taylor, Matthew J. Ballinger, Shaun M. Bowman and Jeremy A. Bruenn

Department of Biological Sciences, The State University of New York at Buffalo, Buffalo, NY, USA

## ABSTRACT

Among eukaryotes with modified nuclear genetic codes, viruses are unknown. However, here we provide evidence of an RNA virus that infects a fungal host (*Scheffersomyces segobiensis*) with a derived nuclear genetic code where CUG codes for serine. The genomic architecture and phylogeny are consistent with infection by a double-stranded RNA virus of the genus *Totivirus*. We provide evidence of past or present infection with totiviruses in five species of yeasts with modified genetic codes. All but one of the CUG codons in the viral genome have been eliminated, suggesting that avoidance of the modified codon was important to viral adaptation. Our mass spectroscopy analysis indicates that a congener of the host species has co-opted and expresses a capsid gene from totiviruses as a cellular protein. Viral avoidance of the host's modified codon and host co-option of a protein from totiviruses suggest that RNA viruses co-evolved with yeasts that underwent a major evolutionary transition from the standard genetic code.

# INTRODUCTION

*Crick (1968)* declared the universal genetic code to be nearly immutable because change would cause 'mistakes' in so many of the proteins of a cellular life form. However, Crick also implied that viruses are a possible exception to this evolutionary 'freezing' process because viruses have but a few protein coding targets. It is now well-established that modified genetic codes have evolved from the universal genetic code at least 34 times (*do Céu Santos & Santos, 2012*). Yet, the consequences of these shifts for virus-host co-evolution remain poorly understood (*Holmes, 2009*; *Shackelton & Holmes, 2008*). Indeed, some authors have proposed that genetic code variants evolved as an antiviral defense (*Holmes, 2009*; *Shackelton & Holmes, 2008*; *Taylor & Bruenn, 2009*). As viruses must use the protein translation machinery of the host, differences in genetic codes could preclude viral transfers among hosts. An evolutionary leap as great as a genetic code change could allow hosts to escape the co-evolutionary struggle with viruses. In agreement with the antiviral hypothesis, viruses are unknown from organisms with modified nuclear genetic codes. Viruses are known to infect the mitochondrial genomes of fungi with alternative mitochondrial genetic codes, but the tremendous divergence of these viruses

Corresponding author
Derek J. Taylor,
djtaylor@buffalo.edu

from known viruses that use the ancestral genetic code obscures their origins (*Shackelton & Holmes, 2008*). Modes of viral adaptation to hosts with non-standard genetic codes remain mysterious.

The CUG codon of the "CTG yeasts" (a diverse monophyletic group that contains human pathogens such as *Candida albicans* and wood-digesting species such as *Scheffersomyces stipitis*) has been reassigned from leucine to serine such that the serine tRNA possesses both derived serine and ancestral leucine sequence motifs (*Butler et al., 2009*; *Santos et al., 2011*). This shift in the genetic code, which replaces a hydrophobic residue with a polar residue, interferes with protein folding and can affect surface residue function (*Feketová et al., 2010*; *Rocha et al., 2011*). As a consequence, CTG yeasts lack the CUG codon from (>90%) functionally relevant positions in proteins (*Rocha et al., 2011*) and rarely receive genes via horizontal transfer compared to fungi with the standard code (*Fitzpatrick, 2012*; *Richards et al., 2011*). Viruses face the same functional barrier posed by this modified code. However, the recent finding of "fossils" of totiviruses in the nuclear genomes of CTG yeast (*Taylor & Bruenn, 2009*) could indicate that exogenous RNA viruses have adapted to a modified nuclear genetic code but remain undetected.

Totiviruses have a double stranded RNA genome (from 4.5–8 kb in size in fungi) and are characterized by an overlapping open reading frame between the capsid gene and the RdRp gene with a programmed ribosomal frameshift. The family (Totiviridae) that contains the totiviruses is ancient with a broad eukaryotic host range and extensive co-evolution in the fungi (*Liu et al., 2012*). Totiviruses 'snatch' the hosts' mRNA caps (modified guanines at the 5' end) with a unique binding mechanism involving at least five proposed residues of the coat protein (*Fujimura & Esteban, 2011*). Frequently, totiviruses in fungi are associated with a satellite killer dsRNA virus that codes for a toxin (*Bostian et al., 1984*). The fossils of totiviruses are best characterized in the CTG yeast, *S. stipitis* (*Frank & Wolfe, 2009*; *Taylor & Bruenn, 2009*). There are four tandem copies of a capsid-like protein gene in the genome of *S. stipitis*, but it is unknown if these copies are translated into proteins. We explored the existence of RNA viruses in the CTG clade of yeasts and their possible mode of co-evolution by attempting to isolate modern and fossil viral genomes and their products from *S. stipitis* and its relatives *Scheffersomyces segobiensis* and *Scheffersomyces coipomoensis*.

## MATERIALS AND METHODS

### Cell cultures

Freeze-dried culture stock was obtained from the USDA ARS Culture Collection for *Scheffersomyces segobiensis* (Santa Maria & Garcia Aser) *Kurtzman (2010)* NRRL Y-11571 (Type strain) and for *Scheffersomyces coipomoensis* (Ramirez & Gonzalez) Urbina & Blackwell, 2012 comb. nov. NRRL Y-17651 (Type strain). Thomas Jeffries (*Jeffries et al., 2007*) provided culture stock for *Scheffersomyces stipitis* strain CBS 6054. Yeast cultures were grown in 150 ml of YPD broth (yeast extract 1%, peptone 2%, and dextrose 2%) with an inoculum (single colony) of cells from streaked YPD agar plates.

### dsRNA assay

Total nucleic acids (depleted of ribosomal RNA) were extracted from whole cells (*Bruenn & Keitz, 1976*). We purified dsRNA using CF-11 chromatography (*Franklin, 1966*). The results (see Fig. S1) were consistent over the two-year period of more than 10 assays, indicating a stable infection.

### Viral particle isolation

The tentative totivirus and satellite virus dsRNAs from *S. segobiensis* were isolated from a CsCl gradient fraction with a density of 1.40 g/cc (Fig. S2), which is expected based on the known *Saccharomyces cerevisiae virus L-A*. Note that even though totiviruses and their satellite viruses are separately encapsidated, there is an overlap in their densities, since particles can encapsidate up to four copies of satellite dsRNA (making a total of about 4.8 kbp, essentially the same as the totivirus at 4.6 kbp).

### PCR, RTPCR, sanger and next generation sequencing

We extracted total RNA from yeast cells using the Masterpure yeast RNA purification kit (Epicentre) and an RNAse-free DNAse treatment. RNA-seq was used to sequence the putative RNA virus, examine the tRNA expression of the host, test for host expression of known fungal viral sequences, and isolate host protein-coding sequences for bioinformatics analysis. Ribosomal RNA species were removed using the Ribo-zero Magnetic Gold kit (Epicentre). We then prepared an RNA-seq library using the ScriptSeq$^{TM}$ v2 RNA-Seq Library Preparation Kit (Epicentre). Libraries were quantified using the Agilent 2100 Bioanalyzer RNA 6000 Pico Chip and submitted to the University at Buffalo Next Generation sequencing facility for RNA sequencing. The facility carried out RNA-seq using 50-cycle paired-end runs on two flow cells of an Illumina HiSeq 2000. *De novo* RNA sequence assembly was carried out in CLC Assembly Cell 4.06 (http://www.clcbio.com) on an Apple Macintosh Mac Pro Xeon 64-bit workstation. The putative viral contigs were reassembled for mapping purposes using the reference assembly algorithm and the *de novo* contigs as references. A total of 409665 reads were mapped to the totivirus with an average coverage of 4404.80 times.

The putative viral sequence was confirmed by Sanger sequencing using random and specific primers for cDNA library construction. We used the Takara 5′-Full RACE Core Set to expand sequence. Sanger and next generation sequences were compared using Geneious version 5.6.3 created by Biomatters (available from http://www.geneious.com/).

Because endogenous RNA viruses of fungi can be fragmented and differ greatly in their nucleotide sequences from known viruses (*Taylor & Bruenn, 2009*), PCR probes alone are often an ineffective tool for paleoviral discovery. We therefore carried out 454 Life Sciences (http://www.454.com) sequencing with GS FLX Titanium series reagents of a DNA library from *Scheffersomyces coipomoensis*. This form of sequencing also permitted multigene bioinformatics analysis of the host protein coding genes. Strain identity of the assembly was confirmed by BLAST analysis (*Altschul et al., 1990*) of the nuclear ribosomal RNA sequences that are known for *S. coipomoensis*. We obtained 614185 reads with about

252 Mb of aligned bases. The assembly carried out in Newbler (http://www.454.com), yielded 14.7 Mb of aligned bases (488 contigs) with an average peak depth of 12X.

To establish that the virus was coded by exogenous RNA and not by the DNA of the host, we compared RT-PCR and PCR products. For RNA templates, DNase-treated extracts were exposed to RT-PCR using the Qiagen one step RT-PCR kit. For DNA templates, nucleic acid extracts were exposed to PCR by excluding reverse transcriptase from the RT-PCR protocol. We amplified a fragment of the single copy xylose reductase gene as a positive control for the PCR of DNA. Primers used were: segoxylF CTGTTCTGAACA-GATCTACCGTGC (xylose reductase), segoxylR AAGTATGGGTGGTGTTCAACTTGC (xylose reductase), SvLgap3F CGCAATACGACCAGGAGATTG (RdRp of virus from *S. segobiensis*), and segoSvLgap3R GTACACCAAGGTTAGTAGACAAG (RdRp of virus from *S. segobiensis*). cDNA synthesis was performed at 48 °C for 30 min, followed by 15 min at 94 °C for reverse transcriptase deactivation and Taq activation. DNA only reactions were added to the thermal cycler 2 min before the end of the previous 94 °C step to activate the Taq polymerase. PCR amplification was done for 35 cycles of 94 °C for 30 s, 48 °C for 30 s, and 72 °C for 1 min. A final extension at 72 °C for 10 min was performed. New sequences from this study have the following Genbank accession numbers: KC610514, KC616419-KC616429.

## Bioinformatics and protein mass spectroscopy

We obtained amino acid sequences from totivirids using the BLAST blastp algorithm with the the capsid gene and the RdRp sequences of *Saccharomyces cerevisiae virus L-BC(La)* as queries and E_values <1e−05. We searched the non-redundant (nr) peptide sequence database (National Center for Biotechnology Information, Bethesda, USA) and the Department of Energy Joint Genome Institute (J.G.I.) genome browser for matches. Fossil or paleoviral copies of Totivirus-like genes were identified by significant BLAST tblastx hits (E_values <1e−05) of relevant NCBI databases using the sequences of *Saccharomyces cerevisiae virus L-BC(La)*. Duplicated capsid gene copies adjacent to the complete integrated viral genomes in the assemblies of *S. stipitis* and *D. hansenii* were assumed to be paralogs (*Taylor & Bruenn, 2009*). These duplicated paleoviruses and closely related (i.e. phylogenetic sister viruses) co-infecting viral strains were omitted for the phylogenetic analyses. Sequences were aligned using MAFFT (*Katoh, Asimenos & Toh, 2009*) with default settings. We carried out maximum likelihood analyses with PhyML 3.0 as implemented in Seaview 4.3.5 (*Anisimova & Gascuel, 2006*; *Gouy, Guindon & Gascuel, 2010*). Model optimization in Prottest (*Abascal, Zardoya & Posada, 2005*) indicated that the LG + invariable sites parameter (*I*)+ gamma parameter for among-site rate variation (G) was the best fit under a Bayesian Information Criterion (BIC). For reliability estimates we used SH-like approximate likelihood ratio tests (*Anisimova & Gascuel, 2006*). Searches were comprised of five random starts under the subtree pruning and regrafting (SPR) algorithm and midpoint rooted.

*Aguileta et al. (2008)* found that concatenation of the two most informative genes from a genomic scale assessment recovered an expected reference fungal phylogeny with strong

support. We used the approach of *Taylor & Bruenn (2009)* who concatenated five of the most phylogenetically informative fungal genes (*Aguileta et al., 2008*) to estimate fungal relations. Accession numbers for the genes used (Minichromosome Maintenance protein 7[MCM7], Kontroller of Growth[KOG1], Elongator complex subunit[ELP3], NAD-specific glutamate dehydrogenase[GDH2], and acetolactate synthase[ILV2]) in the fungal analysis are presented in Table S1. Data were collected from GenBank and from our newly sequenced cultures of *S. coipomoensis* and *S. segobiensis*. Sequences were aligned in MAFFT and exposed to maximum likelihood analyses in RAxML 7.3.2 (*Stamatakis, 2006*) and in PhyML 3.0. Models were partitioned by gene in RAxML using the best-fit models as indicated by PartitionfinderProtein (*Lanfear et al., 2012*).

Relative synonymous codon usage (*Sharp, Tuohy & Mosurski, 1986*) and third position base composition for yeasts and viruses was calculated using the CAIcal server (*Puigbó, Bravo & Garcia-Vallve, 2008*). For viruses the entire open reading frame of the genome was used in the calculations. For yeasts we used the representative genes from the phylogenetic analysis. Species used for the yeast and viral codon usage analyses are listed in Table S2. Bivariate plots of RCSU and base composition were graphed using the R statistical programming language (*Ihaka & Gentleman, 1996*).

FSfinder (*Moon et al., 2004*) was used to locate putative slippery sites and pseudoknots in the totiviral genome. The $tRNA_{CAG}^{Ser}$ for *S. segobiensis* was folded according to the model for *Candida albicans* (*Santos et al., 2011*) using the VARNA secondary structure visualization program (*Darty, Denise & Ponty, 2009*).

Structural information for *Saccharomyces cerevisiae virus L-A* is from the crystal structure of the ScV L-A capsid protein (*Naitow et al., 2002*). For ScV L-BC (La) and SsV L, structural information was predicted by the I-TASSER webserver using ScV L-A cap as a template (*Roy, Kucukural & Zhang, 2010*; *Roy, Yang & Zhang, 2012*).

To examine protein expression of paleoviral copies we isolated crude protein from *S. stipitis* and *S. cerevisiae* by French press and further isolated proteins migrating between 73 kDa and 92 kDa from 10% SDS-PAGE. Protein mass spectroscopy was carried out at the Seattle Biomedical Research Institute Proteomics Core Facility.

## RESULTS AND DISCUSSION

Because the fossil viruses in yeast have a similar architecture to dsRNA totiviruses (*Taylor & Bruenn, 2009*), we carried out a specific chromatographic assay for dsRNA. We detected dsRNA products in *S. segobiensis* with approximately the same gel-estimated size to the totivirus (4.5 kb) and satellite virus of *S. cerevisiae* (1.2 kb, Fig. S1). Viral particles containing both sizes of dsRNAs were also isolated by CsCl equilibrium gradient centrifugation (Fig. S2). No such products were detected in *S. stipitis* or in *S. coipomoensis*. A tblastn using the protein sequences of the two known totiviruses from *S. cerevisiae* as queries revealed significant matches to a contig from a database of our RNA sequence assemblies (extracted from *S. segobiensis* cells) of similar length to the dsRNA on the gel. Assemblies from Sanger sequencing of the RNA virus agreed with the assembly using Illumina RNA sequencing, but the 5′ UTR was complete only in the Illumina assembly. The assembled virus had the

**Peer**J

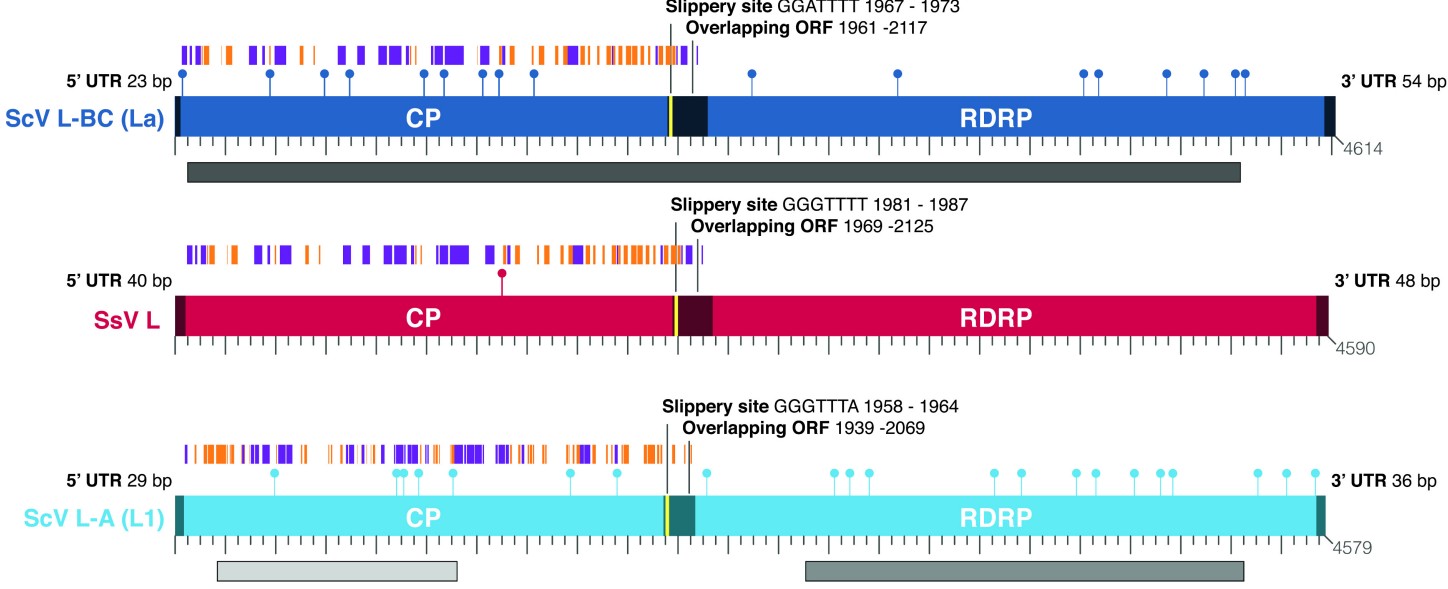

**Figure 1** **Comparison of the genomic architecture of the newly discovered *Scheffersomyces segobiensis virus L* [SsV L] (in red) that uses a modified nuclear genetic code with those of related totiviruses (*Saccharyomyces cerevisiae virus L-BC(La)* [ScV L-BC (La)] and *Saccharomyces cerevisiae virus L-A* [ScV L-A]) that use the standard genetic code.** The figures show overlapping reading frames for capsid (CP) and RNA dependent RNA polymerase genes (RdRp) that are typical of the double-stranded RNA totiviruses. Terminal UTRs (untranslated regions) and central overlapping reading frames are distinguished by dark colored shading. The positions of the ribosomal frameshift sites are indicated in yellow. Coding CUG codons are represented by lollipops. Capsid protein secondary structural domains are shown in purple ($\alpha$-helices) and orange ($\beta$-sheets). BLASTp results for each of the previously known totiviruses to SsV L are shown as gray lines underlying the respective genomes, with darker shades indicating a lower expect value. The scale bar increments represent 50 nucleotides.

genomic architecture of totiviruses with overlapping capsid and RdRp open reading frames flanked by 5′ and 3′ UTRs (Fig. 1). We identified a putative slippery site for ribosomal frameshifting (GGGTTTT) at position 1981 that was independently identified using the frameshift prediction software fsfinder (*Moon et al., 2004*). The five sites identified as functionally important for cap-snatching in totiviruses are conserved in the virus from *S. segobiensis* (Fig. S3), suggesting a cap-snatching mechanism similar to those of well-studied totiviruses. These sites show weak conservation in the fossil copies, consistent with the loss of host mRNA decapping in host-coded elements. The successful PCR amplification of a single copy nuclear gene fragment (xylose reductase gene) from the host genome indicates that the DNA template was of sufficient quality to detect endogenous viral genes using our methods (Fig. S4). However, the primers nested within the viral genome failed to PCR amplify a DNA copy from the host (*S. segobiensis*) genome, but RT-PCR did amplify an RNA copy of the viral gene. The results support the existence of an exogenous RNA virus in *S. segobiensis* with the genomic architecture of a totivirus.

Further evidence of affinity to totiviruses comes from sequence analysis of the virus in *S. segobiensis*. A BLAST blastp analysis of the RdRp-like ORF yielded a conserved domain match ($E = 1.56e - 58$) to RdRP_4, a viral RNA-directed RNA-polymerase family that includes "RdRPs from Luteovirus, Totivirus and Rotavirus". The best expect value ($E = 2e - 137$) obtained was the totivirus, *Saccharomyces cerevisiae virus L-BC(La)*,

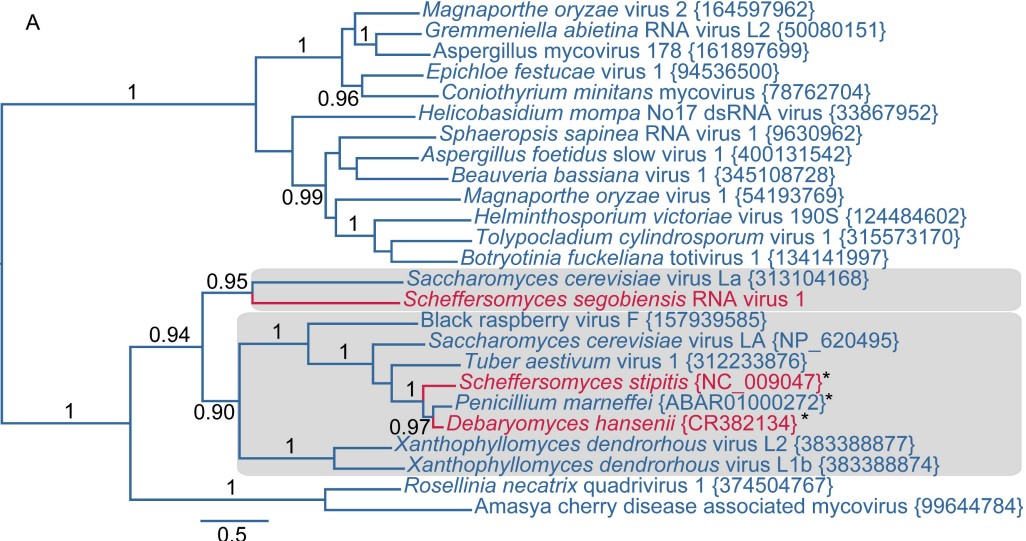

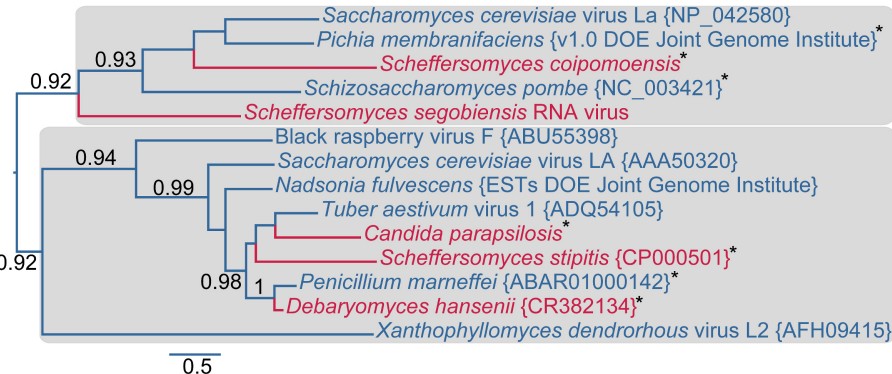

**Figure 2 Evolutionary relationships of exogenous and endogenous totiviruses showing the derived and non-monophyletic positions of viruses and paleoviruses from CTG yeast (in red).** Asterisks indicate paleoviral sequences. Numbers are support values (SH-like approximate likelihood tests estimated in PhyML 3.0). The phylograms are estimated using the maximum likelihood (ML) optimality criterion from alignments of predicted amino acid residues of (A) the RNA dependent RNA polymerase gene (RdRp) and (B) the capsid gene. Shaded boxes indicate the two major clades of totivirus-like sequences. Genbank Accession numbers are provided in parentheses.

with an identity of 37% of residues. The RdRp gene phylogeny (Fig. 2A) positioned the tentative species *Scheffersomyces segobiensis virus L* within the totiviruses and most closely related to *Saccharomyces cerevisiae virus L-BC(La)*. Support values are strong enough to rule out random error as an explanation for the evolutionary position of *Scheffersomyces segobiensis virus L* within the totiviruses. The less conserved capsid gene tree (Fig. 2B) showed a similar association, but with fewer outgroup sequences and more paleoviruses. We detected fossil totiviruses in the genomes of *S. coipomoensis*, and *Pichia membranifaciens* and a putative totivirus in the assembly of *Nadsonia fulvescens*. We deem

the sequences from *Nadsonia* to be putative viral sequences because they are present in the RNA-based EST libraries, but not the DNA based genome assembly (see *Liu et al. (2012)* for a discussion of this approach to dsRNA virus discovery).

As with other totiviruses, we found evidence that *Scheffersomyces segobiensis virus L* has a putative killer satellite virus. We carried out a BLAST search to identify candidate RNA contigs for the dsRNA band observed earlier. A sequence was obtained of the correct size (1.2 kbp) that had a significant match to the killer satellite K2 virus of *S. cerevisiae*. An RTPCR with specific primers confirmed that the contig was not coded in the DNA genome of the yeast. The existence of a satellite virus bolsters the evidence for a totivirus in the CTG clade.

The evolutionary and genomic evidence indicates that *Scheffersomyces segobiensis virus L* originated from exogenous totiviruses that use a standard genetic code. The RdRp permits the deepest assessment of evolution, and it reveals a derived position of the *Scheffersomyces segobiensis virus L* at the tip of the standard code totivirids. The known paleoviruses in the CTG clade are distantly related to *Scheffersomyces segobiensis virus L* and lack the ability to produce the fusion gene product typical of totiviruses (*Taylor & Bruenn, 2009*). Nor is there any evidence of a functioning totivirus genome being coded in the DNA from known genomes of yeasts. Moreover, our RTPCR results indicate that the DNA genome of the host, *S. segobiensis*, lacks coding sequences related to the viral genome. Finally, successful endogenization of a virus that imparts a selective disadvantage by obligately decapping host mRNA seems unlikely. Most of the successful paleoviruses in the CTG clade possess functionally differing residues at the decapping sites. Because the CTG virus we discovered has the conserved residues for such a decapping mechanism, an "escaped" genome hypothesis requires that the unique decapping mechanism was lost in the host genome and then re-evolved in the escaped viral genome – also unlikely. We conclude that adaptation to the genetic code shift of hosts happened in exogenous viruses.

Our viral evolutionary trees are consistent with the jumping of viruses between hosts with different genetic codes. Host products from the CTG clade are phylogenetically interspersed with standard code sequences in both major clades and the virus/paleovirus phylogenies bear little resemblance to the host relationships. For example, a virus from the truffle (*Tuber aestivum*) is most closely related to the fossil virus in the CTG yeast *S. stipitis* rather than the virus of another genus of basidiomycete, *Xanthophyllomyces*. Horizontal transfer of paleoviruses could be a source of some evolutionary noise, but this process appears rare in the CTG clade. The timescale of the totivirus evolution is difficult to estimate because of the dearth of reliable fossil calibrations for fungi (*Berbee & Taylor, 2010*; *Rolland & Dujon, 2011*). However, our observed close sequence and structural similarity for rapidly evolving capsid proteins of RNA viruses likely postdates the ancient split of ascomycetes with basidiomycetes (452 MYA to 1400 MYA) and the origin of the CTG clade (>150 MYA) (*Berbee & Taylor, 2010*; *Massey et al., 2003*; *Pesole et al., 1995*).

The evolutionary position, CUG codon usage, and tRNA expression evidence are consistent with the host, *S. segobiensis*, having a modified CTG genetic code. Our phylogenetic analysis (Fig. 3A) of nuclear protein coding genes revealed strong support

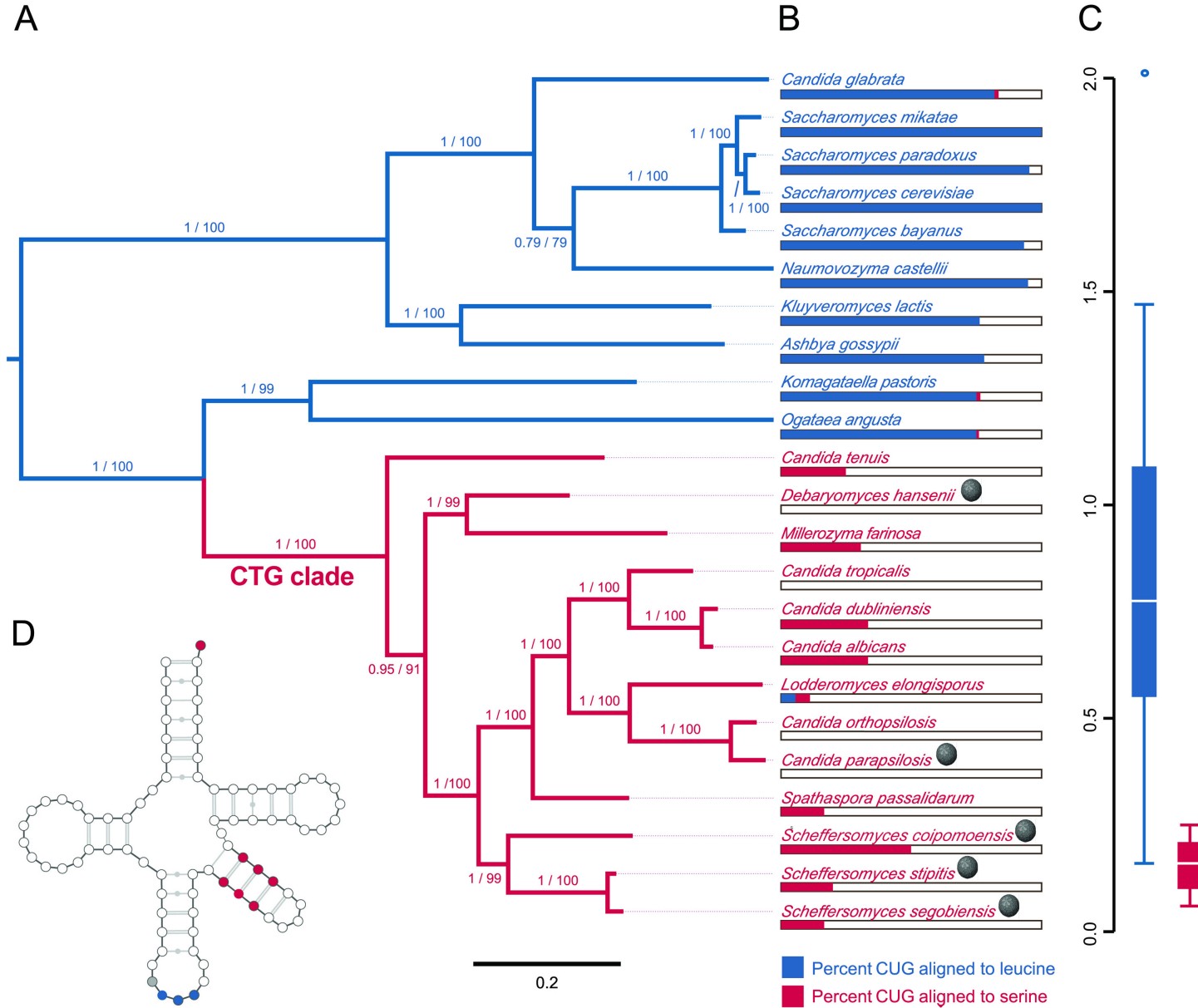

**Figure 3 Evidence that the viral host, *Scheffersomyces segobiensis*, uses the modified genetic code of the "CTG" clade.** (A) Midpoint-rooted maximum likelihood phylogram of CTG clade yeasts and related yeasts based on the protein sequences of the five most phylogenetically informative genes for fungi. Branches are labeled with support values from approximate likelihood ratio tests and nonparametric bootstrapping. Blue shading indicates standard code yeasts, while red shading indicates CTG code yeasts. Gray spheres indicate lineages with evidence of past or prior infections with totiviruses. (B) Comparison of CUG codon positions in each taxon versus homologous amino acid residues in *Saccharyomyces cerevisiae*. Blue bars represent the percent of *S. cerevisiae* leucine residues that are coded by the CUG codon for each taxon, while red bars represent the percent of *S. cerevisiae* serine residues that are coded by CUG for each taxon. (C) Boxplots showing relative synonymous codon usage (RSCU) for CUG codon usage by each taxon, the blue plot represents CUG RSCU values for non-CTG clade yeast, while the red plot represents CTG clade yeast including *S. segobiensis*. (D) The secondary structure model of the CTG clade type of tRNASER detected in *S. segobiensis*. The red shading indicates serine identifier sites, while blue shading indicates standard leucine identifier sites. The gray site is a typical guanine residue of the tRNASER of CTG yeast that lowers the leucine amino-acylation efficiency.

for the placement of the host yeast within the CTG clade and as a sister species to the CTG species *S. stipitis*. The monophyly of the CTG clade is well established in evolutionary genomics (*Butler et al., 2009*; *Louis et al., 2012*; *Wohlbach et al., 2011*). The close sister group relationship of *S. segobiensis* and *S. stipitis* has also been independently supported by several studies using the ribosomal rRNA gene family (*Cadete et al., 2012*; *Kurtzman, 2010*; *Urbina & Blackwell, 2012*). CUG comprises a substantial percentage of codons for leucine residues in standard code yeasts, while being almost absent in the CTG yeasts at the homologous leucine position (Fig. 3B). The usage of CUG in CTG yeast (including *S. segobiensis*) is underrepresented compared to the standard code yeast (Fig. 3C). A BLAST search of RNA contigs for the characteristic modified chimeric serine tRNA of *S. stipitis* that recognizes CUG revealed a significant match in *S. segobiensis*. That is, the sequence has both serine and leucine identity sites (Fig. 3D). We failed to detect a "standard code" leucine tRNA. The pre-tRNA species (Fig. S5) had unique mutations in the flanking regions from *S. stipitis*, consistent with the modest divergence of a sister species.

As the genetic code shift had a profound functional effect on the proteins of yeasts, we expected the virus to adapt to the shift. We found that *Scheffersomyces segobiensis virus L* had but a single codon of the modified CUG type. This evolutionary loss of CUG codons resulted in the lowest CUG frequency known among related mycoviruses, where the CUG codon is generally overrepresented. The sole CUG in *Scheffersomyces segobiensis virus L* occurs at a position in the capsid protein that appears to be structurally unimportant (Fig. 1). *Scheffersomyces segobiensis virus L* appears to have adapted to the host shift in genetic code by eliminating functionally relevant CUG codons.

In plots of relative synonymous codon usage (RSCU) versus third position base composition (Fig. 4A; Table S2), *Scheffersomyces segobiensis virus L* grouped with CTG yeasts rather than with other totivirids. The same pattern of *Scheffersomyces segobiensis virus L* grouping with CTG yeasts to the exclusion of other viruses was found for relevant leucine codons (Fig. 3B, C). In *S. cerevisiae*, CUN codons are decoded by either tRNA-UAG or by tRNA-GAG. But in the CTG clade yeast, CUN codons are decoded differently. Here a derived tRNA-CAG decodes the reassigned CUG codon, while the remaining CUN family codons are decoded by a single tRNA-IAG. The inosine interacts only weakly with CUA compared to CUC and CUU in *C. albicans* (*Massey et al., 2003*), and seems the most likely driving force behind reduction in CUA usage in CTG yeasts. We note that *Scheffersomyces segobiensis virus L* lacks a bias in base composition at the third position for the examined codons, suggesting that the observed codon usage bias in *Scheffersomyces segobiensis virus L* is more complicated than base compositional shifts alone.

Our analysis of the genome of *Scheffersomyces segobiensis virus L* is consistent with adaptation to offset the functional effects of the genetic code shift in the host. But, our results also indicate that the endogenization of viral genes by host yeasts of both genetic codes is more common than previously thought. Presently, it is unknown if these endogenous non-retroviral genes function as proteins or are merely transcriptional noise. Here, we show that at least one of these genes derived from a totivirus is expressed as a protein. Isolation of proteins migrating between 73 kDa and 92 kDa from *S. stipitis* yielded

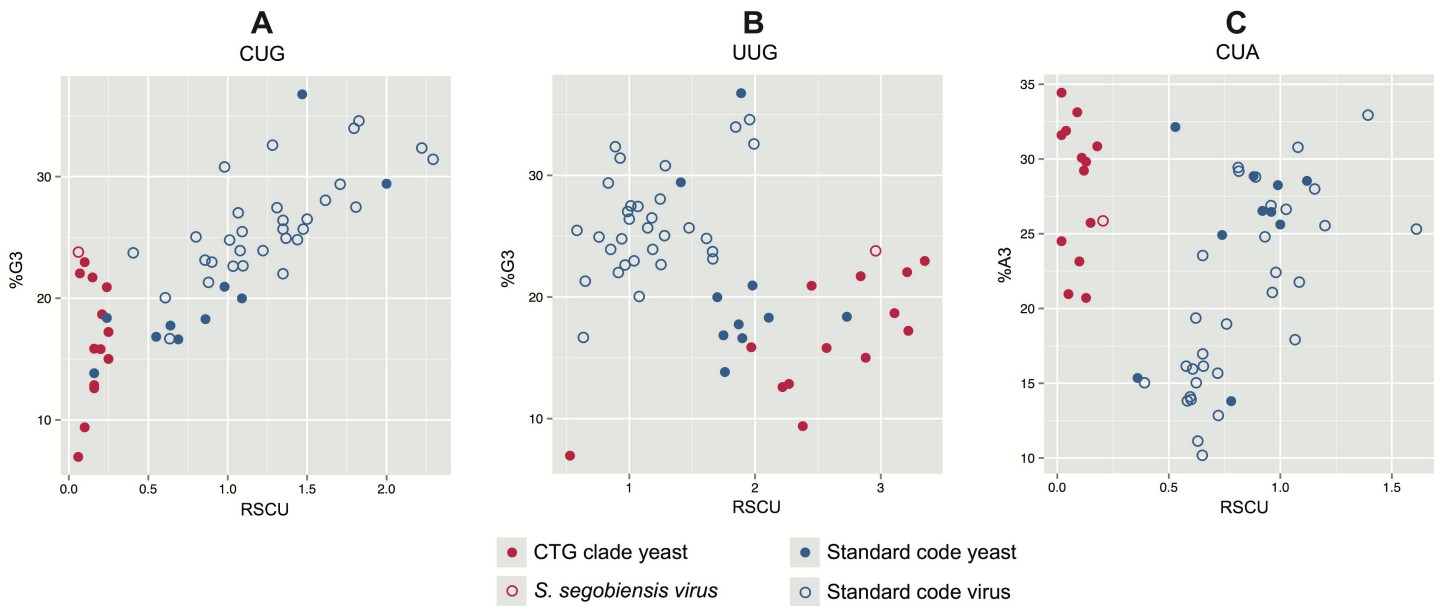

**Figure 4 Bivariate plots of relative synonymous codon usage (RSCU) for serine and leucine versus third position base composition in yeast and their dsRNA viruses. CTG clade yeasts are shown as solid red points, and their viruses as hollow red points.** Standard code yeasts are shown as solid blue points and their viruses as hollow blue points. (A) CUG is used by standard code yeasts and their viruses but avoided by CTG clade yeasts and *Scheffersomyces segobiensis virus L*. (B) Leucine codon UUG is overused by CTG clade yeasts and *S. segobiensis virus L* relative to standard code yeasts and their viruses. (C) Leucine codon CUA is used by standard code yeasts and their viruses, but avoided by CTG clade yeasts and *Scheffersomyces segobiensis virus L*. Comparison with %N3 suggests that these codon preferences are not attributable to nucleotide composition alone.

approximately 535 proteins, as estimated from the equivalent size range of proteins in *S. cerevisiae*. The mass spectroscopy analysis was able to unambiguously identify the capsid protein (cap) of *Saccharomyces cerevisiae virus L-A* in the *S. cerevisiae* control and 153 proteins in *S. stipitis*. By this method, we were able to detect one of the four Non-retroviral integrated RNA virus (NIRV) capsid polypeptides from *S. stipitis*. Figures S6A and S6B show the distribution of tryptic peptides detected by mass spectroscopy from *S. stipitis* capsid4 and *Saccharomyces cerevisiae virus L-A* cap. All of the *S. stipitis* virus-like peptides had probabilities of 0.999 or greater. Our inability to detect the remaining cap proteins from *S. stipitis* may be due to the lack of sensitivity of the method used. We were able to detect a single tryptic peptide from *Saccharomyces cerevisiae virus L-BC(La)* cap, which is present at about one tenth of the concentration of *Saccharomyces cerevisiae virus L-A* cap. Our results indicate that the co-option of a non-retroviral RNA viral protein has occurred as with retroviral proteins (e.g. syncytin genes *Feschotte & Gilbert, 2012*). Although expression of similar capsid proteins has an antiviral effect (*Valle & Wickner, 1993*; *Yao & Bruenn, 1995*), the current function of the co-opted viral proteins in yeast is unknown.

## CONCLUSIONS

Our discoveries indicate that a major evolutionary transition involving a change in the genetic code of the fungi failed to result in permanent host escape from viruses. We found evidence of present or past viral infection in five lineages of yeasts with a modified genetic

code. Thus, viral infection is likely widespread in the CTG clade of fungi. The mode of viral adaptation recalls the prediction of *Crick (1968)* that some viruses would be less susceptible to evolutionary "freezing" because they present a reduced protein target. The genomes of totiviruses are among the smallest known for RNA viruses (*Holmes, 2009*). Still, even with a small viral genome we found evidence that exogenous viral adaptation was associated with the elimination of modified codons from functional positions. Our results also highlight the value of recent paleovirological approaches to understanding virus-host biology (*Aswad & Katzourakis, 2012*; *Feschotte & Gilbert, 2012*; *Holmes, 2011*; *Koonin, 2010*; *Patel, Emerman & Malik, 2011*). Fossil copies in the genomes of yeasts informed us about prior infections and made possible our discovery of a virus adapted to a modified nuclear genetic code. Moreover, we found that at least one of these "fossil" genes is co-opted by the yeast host and expressed as a protein.

## ACKNOWLEDGEMENTS

We thank Thomas W. Jeffries for *Scheffersomyces stipitis* strain CBS 6054. Jennifer Jamison (Next-Generation Sequencing and Expression Analysis Core, State University of New York at Buffalo) and Yuko Ogata (Proteomics Shared Resource, Fred Hutchinson Cancer Research Center) provided technical advice.

### Funding

Sequencing and material support was provided by the University at Buffalo. The funders had no role in study design, data collection and analysis, decision to publish, or preparation of the manuscript.

### Competing Interests

The authors declare no competing interests.

### Author Contributions

- Derek J. Taylor, Matthew J. Ballinger and Jeremy A. Bruenn conceived and designed the experiments, performed the experiments, analyzed the data, contributed reagents/materials/analysis tools, wrote the paper.
- Shaun M. Bowman conceived and designed the experiments, performed the experiments, contributed reagents/materials/analysis tools.

### DNA Deposition

The following information was supplied regarding the deposition of DNA sequences: Eleven GenBank accession numbers are based on the present manuscript: KC610514, KC616419-KC616429

### Supplemental Information

Supplemental information for this article can be found online at http://dx.doi.org/10.7717/peerj.50.

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
