# Peer review of "Virus-host co-evolution under a modified nuclear genetic code"

_PeerJ, doi:10.7717/peerj.50_

## Round 0.1 · original submission · Minor Revisions

The reviews received both agreed that the data presented was suitable; however, there was a bit of disagreement on the way in which it was presented. I do agree with some of the recommendations which may need to be addressed prior to publication. The abstract should be adapted to better suit the message you want to present. Some of the comments by reviewer 1 about adjustment of the Materials and Methods subheadings does appear to be valid to address clearly the methods used.

I do feel this work will have an impact on how we view genome interactions considering the breadth of genome information we are now able to collect.

Below are some specific suggestions to ease the reading of the manuscript:

Example of annotation: /PREVIOUS FORM/SUGGESTED FORM/(ADDITIONAL NOTE).

SUGGESTION NOTES:

Page 4:
80: /De novo/de novo/.
78: CLC (website citation).

Page 5:
84: Geneious (website citation).
85: /454 Life Sciences/Roche 454 Life Sciences/(website citation).
87: (BLAST citation) needed because of extensive use further on.
101: Have Genebank accession numbers been obtained yet?
There is a release when published option for Genbank submissions.
102: /blastp/BLAST blastp/.
103: /cut-off of E<0.05/cutoff E_value of E<0.05/.
104: /nr peptide sequence database/non-redundant 'nr' peptide sequence database/.
107: /tBLASTx/BLAST tblastx/

Page 6:
125-127: (Are there identifiers for these encoded PEPTIDES or genes?)

Page 7:
138: /The tRNA was folded using VARNA secondary structure visualization
program/ The folding of tRNA was predicted using the VARNA secondary structure visualization
program/.

Page 8:
156: /tblastn/BLAST tblastn search/; be consistent in usage of BLAST notations
(tblastn program option in this case).
174: /blastp/BLAST blastp/.
176: /greatest expect value (E= 2e-137)
/highest expect value (E= 2e-137)
obtained/.

Page 9:
186: /EST libraries but/EST libraries, but/

Page 10:
216: /another basidiomycete,/another genera of basidomycete,/

Page 12:
265: /Do these endogenous non-retroviral genes function as
proteins or are they merely transcriptional noise?
/It is not clear whether these endogenous non-retroviral genes produce any functional protein or are just expressed at the transcriptome level./.

Figure 1:
/BLASTp/BLAST blastp/

Page 14:
310: (There are no PUBLISHER or PAGES listed).

Page 16:
372: /Bmc Bioinformatics/BMC Bioinformatics/(There are no PAGES listed).

Page 18:
409: /Bmc Biology/BMC Biology/.

·

Basic reporting

A very clear, straightforward, well written article.

Experimental design

The article employs well-established techniques, and I see no problems with the way these methods are used.

Validity of the findings

The results appear valid and are appropriately supported by the data.

Additional comments

As the first report of virus coevolution with a host that uses a modified genetic code, this certainly is a very interesting and important article.

Reviewer 2 ·

Basic reporting

The manuscript has lot of grammatical errors all over it. This makes proofreading essential before the submission. There are more than 100 first person plural narratives in the text which need to be changed to third person passives.
The abstract is not a good summary of the work and it doesn't show why the authors are claiming that the RNA virus is co-evolved with the host.
The material and methods needs some changes in the subheadings (like PCR, RT-PCR and DNA sequencing subheading that explain more the Next generation sequencing and RNA sequencing rather than DNA sequencing). Therefore the materials and methods section should be restructured again. There are some occasions that the reference or the name of the company is missed for a device, software or a kit.
The result is vague and it is not clear why the authors used different molecular nalysis to confirm their hypothesis. (there used different technologies and molecular analysis in this research but it is not clear for what reason this has been done.). For example I couldn't understand using both RNA seq and also RT-PCR. Was the second one a confirmation or what? Or It is not clear why both 454 FLX sequencing was done for the whole yeast genome when we needed just a confirmation of existance of some Totivirus genes in the nuclear genome. I believe there are some unnecessary works that is done without any particular reason or the result is not fully included in this manuscript for other reasons.
There is not enough discussion about results and the references need a few corrections like BMC Genomics which is written in small letters as Bmc genomics..

Experimental design

In the materials and methods (line 91) and result (line 168) the authors used RT-PCR to find out the nuclear genomic copy of the virus. I believe they have mistaken ordinary PCR with RT-PCR which is for RNA amplification not genomic DNA. I am not sure why they have not used any RNAase prior to PCR to be sure about exogenous virus RNA degradation.
It is not fully described why different high throughput molecular analysis (like RNA seq (illumina), DNA seq (454), mass spectroscopy was used for this purpose. Therfore it is really hard to follow the results in this manuscript.

Validity of the findings

I believe the authors have validated their result well although for some part of their result they could have a better discussion which is missing in this manuscript.

Additional comments

The study was quit interesting but the way it was presented takes the positive energy out from the reader.lack of a smooth flow of the results and enough discussion make the manuscript hard to be followed. I found lots of grammatical mistakes and simple writing errors all over the manuscript which seems to be easy to avoid prior to submission. Using a number of first person plurals is not common in a research manuscript, therefore changing them to third person passive would be appropriate. for the error in the text I am going to give you some examples but there are lots of same mistakes which needs attention:
For BLASTp or blastp be consistent. in line 138 change the sentence for example " The folding of tRNA was predicted by using VARNA secondary structure visualization program". Change the De novo to de novo in line 80. in line 156 use "A tblastn search using ..... . in line 176 use "The highest expected value". or finally in line 256 change the sentence to " It is not clear whether these endogenous non-retroviral genes produce any functional protein or just expressed in transcriptome level."
The abstract is not a good summary of the work and it doesn't show the reason for claiming that the RNA virus is co-evolved with the host.
The material and methods needs some changes in the subtitles (like PCR, RT-PCR and DNA sequencing that explain more the Next generation sequencing and RNA sequencing rather than DNA sequencing). Therefore the material and method section should be restructured again. For example in the bioinformatic section, the protein extraction was describes!
There are some occasions that the reference or the name of the company is missed for a device, software or a kit. for example line 84 is missing the reference for Generious software. or line 89 for Newbler
The result is vague and it is not clear why the authors used different strategies for their purpose. (there are different technologies and molecular analysis used in this research but it is not clear for what reason these have been done.). for example I couldn't understand using both RNA seq and also RT-PCR in the genome) was the second one a confirmation or what? Or It is not clear why both 454 FLX sequencing was done for the whole yeast genome when we needed just a confirmation of some Totivirus in the nuclear genome.
In the materials and methods (line 91) and result (line 168) the authors used RT-PCR to find out the nuclear genomic copy of the virus. I believe you have mistaken ordinary PCR with RT-PCR which is for RNA amplification not genomic DNA. I am not sure why you have not used any RNAase prior to PCR to be sure about exogenous virus RNA degradation.
There is not enough discussion about results and the references need a few corrections like BMC Genomics which is written in small letters as Bmc genomics..

Hope these comments would be helpful and good luck with your research.

.

---

## Round 0.2 · accepted · Accept

I think most of the points brought up in the initial review were addressed and now feel the manuscript is in good shape for publication. I thank you for your contribution and your efforts to help expand the knowledge in co-evolution using this virus-host interaction system. Perhaps this work may serve as a model for further studies in co-evolution interactions.